# Risk Factors Associated with Feeding Children under 2 Years in Rural Malawi—A Formative Study

**DOI:** 10.3390/ijerph16122146

**Published:** 2019-06-17

**Authors:** Kondwani Chidziwisano, Elizabeth Tilley, Rossanie Malolo, Save Kumwenda, Janelisa Musaya, Tracy Morse

**Affiliations:** 1Centre for Water, Sanitation, Health and Appropriate Technology Development (WASHTED), Polytechnic, University of Malawi, Private Bag 303, Chichiri, Blantyre 3, Malawi; rossaniedaudi@yahoo.com (R.M.); skumwenda@poly.ac.mw (S.K.); 2Department of Environmental Health, Polytechnic, University of Malawi, Private Bag 303, Chichiri, Blantyre 3, Malawi; Elizabeth.Tilley@eawag.ch; 3Department of Civil and Environmental Engineering, University of Strathclyde, Level 5 James Weir Building, G1 1XQ, Glasgow, UK; tracy.thomson@strath.ac.uk; 4EAWAG: Swiss Federal Institute of Aquatic Science and Technology, 8600 Dübendorf, Switzerland; 5Department of Pathology, College of Medicine, University of Malawi, P.O Box 360, Chichiri, Blantyre 3, Malawi; jmusaya@medcol.mw

**Keywords:** food hygiene, food safety, complementary food, child feeding, Malawi

## Abstract

Diarrhoeal disease remains one of the leading causes of morbidity and mortality in the under-five population, particularly in low income settings such as sub-Saharan Africa. Despite significant progress in sanitation and water access, faecal-oral infections persist in these populations. Therefore, a better understanding of these transmission pathways, and how potential risk factors can be reduced within low income contexts is needed. This study, conducted in Southern Malawi from June to October 2017, used a mixed methods approach to collect data from household surveys (*n* = 323), checklists (*n* = 31), structured observations (*n* = 80), and microbiological food samples (*n* = 20). Results showed that food prepared for immediate consumption (primarily porridge for children) posed a low health risk. Poor hygiene practices increased the risk of contamination from shared family meals. Faecal and Staphylococcal bacteria were associated with poor hand hygiene and unhygienic eating conditions. Leftover food storage and inadequate pre-consumption heating increased the risk of contamination. Improvements in food hygiene and hand hygiene practices at critical points could reduce the risk of diarrhoeal disease for children under 2 years but must consider the contextual structural barriers to improved practice like access to handwashing facilities, soap, food and water storage.

## 1. Introduction

Diarrhoeal disease remains one of the leading causes of morbidity and mortality in the under-five population globally, with approximately 424,000 deaths annually [1]. The Malawi Demographic and Health Survey (2016) indicated that 22% of children under the age of five years had diarrhoea two weeks before the survey, a slight increase from the 17.5% reported in 2010 [2,3]. The high prevalence of childhood diarrhoea could be one of the contributing factors to the high under-five mortality rate (62 deaths per 1000 births) experienced in Malawi [2]. Primary sources of direct and indirect contamination have been clearly outlined in the F-diagram for decades [4], highlighting the key transmission routes for pathogenic organisms. Recent research undertaken in low income countries [5,6,7,8,9] has expanded on the F-diagram to better illustrate the links between under-five behaviours, daily activities, and faecal exposure. Several studies have now reported the significance of child play areas, mouthing, geophagia, animal contact, and water as potential sources of diarrhoeal disease transmission within these settings [10,11,12,13,14].

Previous studies have highlighted the important role of food hygiene in diarrhoeal disease prevention, a key but often neglected area of the F-diagram. However, significant numbers of pathogens have been isolated in complementary food in Sub-Saharan Africa, Bangladesh and Peru [15,16,17,18], most of which have been associated with prolonged food storage at high ambient temperature, seasonality, and unclean utensils [16,17,18,19,20,21,22]. In addition, studies have reported significant associations between diarrhoeal disease and lack of a kitchen, kitchen cleanliness, handwashing at critical times, feeding practices, waste disposal and storage of food on the floor [23,24,25,26,27,28]. Post-cooking activities that include improper handling of kitchen utensils, and poor handwashing practices are risk factors that have been associated with diarrhoea-causing pathogens in food in Malawi [29,30].

Diarrhoeal disease interventions have traditionally focused on water, sanitation and handwashing with soap (WASH), with little integration of food hygiene programmes [12]. Nevertheless, the contribution of food in the transmission of diarrhoeal disease has been clearly outlined by a 2015 WHO report which attributed 70% of the burden of foodborne disease to sub-Saharan African and South East Asia, with 40% affecting children under the age of five [31]. Despite rising evidence of the role of food in disease transmission, attempts to model the complex mechanisms which potentially link these to diarrhoeal disease, enteric enteropathy, under nutrition and child development are limited, primarily due to the myriad contributing factors [32,33,34,35,36]. Recent studies conducted in Nepal, Gambia and Mali have shown the potential impact of child caregiver training, follow-up and participatory approaches (including hazard analysis principles at household level) on the safety of domestically produced complementary foods [37,38,39,40]. Previous studies on diarrhoeal disease prevention conducted in Malawi indicated the importance of handwashing, water treatment and use of latrine in diarrhoea prevention [41,42]. However, few studies have explored child feeding practices and their potential effects on childhood diarrhoea in this setting.

Research has shown the need to apply the Hazard Analysis Critical Control Point (HACCP) strategy to identify hazards associated with complementary food preparation, handling, storage and child feeding practices, with subsequent identification of effective control points [19]. Although previous studies have contributed to our understanding of complementary foods as a source of pathogen transmission, most have focused specifically on the levels of microbial contamination in foods. There is still a need to understand cultural practices including how, when and what children under five are fed throughout the day, the other items they are mouthing, behavioural factors that contribute to caregiver actions, and the microbiological quality of foods provided across that time span. By assessing these potentially risky practices, we can assess cross-cultural similarities with other studies in the region, and provide a basis for developing effective interventions both regionally and locally to improve food hygiene practices. Therefore, the specific objectives of this paper were to: (1) identify practices and associated factors at household level related to food contamination, child mouthing, handwashing practices and kitchen utensils; and (2) develop a flow diagram of the preparation, storage and feeding of main complementary foods with the aim of understanding the local context in which child feeding, food preparation and storage take place. This study was a component of a larger body of work to understand potential infection pathways of children under the age of five years in an intervention trial to improve child health in rural settings of Malawi.

## 2. Materials and Methods

### 2.1. Setting and Population

This was a formative study undertaken in the Southern Region of Malawi in Chikwawa District from June to October 2017. Covering an area of 4878 km^2^, the district has a population of 564,684, of which 16% are under the age of 5 years [2,43]. Full vaccination coverage is 76.4%, which is in line with national coverage (75.8%). Acute respiratory infections among under-5 children were 6% (5% nationally). 70% of children under 6 months were reported to be exclusively breastfed with 88.6% being introduced to solid foods after the recommended 6 months. Being rural, the Chikwawa district is one of the districts with the lowest literacy rate (65.2% young female and 70.4% young male) and ranks low on the wealth index indicators [2,44]. Access to improved water sources in Chikwawa is 86.6%. However, improved sanitation coverage is 42.4% [2]. Twenty-four percent of households have handwashing facilities, which is slightly higher than at the national level (19.5%). However, only 10.7% of households have handwashing facilities with soap and water, despite 44% of households having soap available for other needs within the home [2,44].

### 2.2. Recruitment and Participants

Malawi is divided into 28 Districts, which are subdivided into Traditional Authorities (TAs). Each TA contains villages, which are administered by chiefs and/or village heads. There are 12 Traditional Authorities (TAs) within Chikwawa district. This work was based in two TAs selected in collaboration with the District Health Office.

The number of households and population in the study area were obtained from the community health workers’ (locally known Health Surveillance Assistants) register. Households were selected in the 4 stages of the study using systematic random sampling from the register. All participants in Stages 1, 2, and 4 were part of Stage 3 (Table 1). A sample size of 295 was calculated based on the Chikwawa district diarrhoea prevalence of 26.3%, with an acceptable error margin of 5% [44]. Taking into account non-responses and missing data, the sample size increased to 323.

To ensure that there were no significant variations in access to water, all recruited households resided within a 500 m radius of a functioning protected borehole. Eligible households had a child aged between 3 and 24 months. The age of the children was verified through birth and/or immunization records supplied by the caregiver. Physical recruitment was conducted by trained research assistants with the approval and support of community health workers (Health Surveillance Assistants) and traditional leaders (village chiefs). Written consent was received from all households willing to participate before allocation of a household identification number and associated barcode. Pre-testing of all data collection tools was conducted to identify and eliminate irrelevant questions while key questions were further edited for easy understanding.

### 2.3. Observations

To identify critical control points for subsequent microbiological sampling, checklist and structured observations followed by in-depth interviews were used. Initially, checklist observations were conducted in 31 households that were selected from the list of recruited 323 households using systematic random sampling to identify a list of behaviours that were considered to put children at risk of developing diarrhoea. For the checklist observations, a household was visited over two consecutive days: 6 am–12 pm on the first day and from 12–6 pm on the second. The aim was to capture all events of interest that occurred in a day, including child mouthing (geophagia inclusive), practices around food storage, preparation and feeding/eating. In addition, the child caregiver’s handwashing practices at critical times were observed: before food preparation, before child feeding/eating, after toilet use and after cleaning a child following defecation. “Child feeding practices” in this paper refers to complementary food given to the child after 6 months, including child self-feeding, while “child caregivers” include any household members, including parents, who are responsible for the daily care of the targeted child. Responsibilities of the caregiver include feeding and preparing the child’s food, bathing, and assisting the child during defecation. Subsequent structured observations were conducted, specifically focusing on behaviours noted during checklist observations. In total, 80 households were targeted for structured observations (including those previously used for checklist observations) and visited once between 6 am and 1 pm. Checklist observations had indicated that the majority of food preparation and feeding events took place in the morning.

In-depth interviews followed each structured observation period to understand how and why some practices were conducted as observed. To ensure good quality data, debriefing sessions were conducted daily where supervisors and enumerators cross-checked observation forms to ensure that data were complete and consistent in reporting observed practices.

A team of 5 female observers (BSc holders in Social Sciences (*n* = 1) and Environmental Health (*n* = 4)) were trained to conduct in-depth interviews, checklist and structured observations. The training package included details of the research study, the theoretical science of observational research (Hawthorne effect inclusive) and observation tools. The research team opted for female research observers since child care at community level in Malawi is primarily performed by females.

### 2.4. Demographic, Socio-Economic and Hygiene Proxy Questionnaire

Following the observation stage, a structured questionnaire was conducted which contained closed questions and captured demographics, hygiene behaviours, child health status, and socio-economic proxy measures. At the end of the interview with the child’s primary caregiver, enumerators conducted spot checks and recorded hygiene proxy measures such as the presence and condition of the latrine, the presence, location and type of handwashing facilities (including the availability of soap and water), the presence of a kitchen, and the presence of animals and their faeces. Face-to-face interviews were conducted in Chichewa, the local language of Chikwawa district. Behavioural factors for each of the critical areas were assessed using the RANAS (Risk, Attitude, Norm, Ability and Self–regulation) model [45,46]. The structured questionnaire was conducted by ten well-trained and experienced research assistants who were fluent in Chichewa.

### 2.5. Microbiological Sampling and Analysis

In total, 224 microbiological samples were collected over a 24 h period in 20 households selected from the list of 323 recruited households using systematic random sampling to assess the extent of bacterial contamination in foods consumed by target children. Households were visited on three occasions within 24 h for sampling, as outlined in Figure 1.

All sampling points in the study were informed by the observations which were conducted prior to sampling. The child’s most frequently consumed foods were sampled for microbiological analysis. Sampled foods included the morning porridge, and nsima, which is the main meal for lunch and dinner. Both porridge and nsima are common complementary foods in Malawi prepared locally at home from maize, millet or sorghum flour. For porridge, the liquid is cooked for 30 min before adding sugar, salt, pounded ground nuts or milk depending on availability. The porridge was given to children after 10–15 min of cooling. Samples of porridge were taken in the morning after cooking and after serving (Figure 1). Porridge was served in a plate and eaten with a spoon, child’s hands (self-feeding) or caregiver’s hands.

Nsima is prepared just like porridge; however, more maize flour is added to produce a thicker consistency, which is cooked for approximately 40 min, and no other ingredients are added. Nsima is prepared for immediate consumption. However, we occasionally observed that it was kept longer, e.g., overnight, to be eaten the following morning. As such, nsima samples were taken during dinner after cooking, and the following morning from leftovers of nsima eaten during dinner (Figure 1). Nsima is served in a plate and eaten using hands with a relish (Scheme 1).

Relish is the word used to describe the side dish that is served with nsima. The most common relishes recorded in the study were beans, vegetables and fish. Depending on availability, tomatoes, onions, salt and cooking oil were added to the relish and cooked together. The relish is cooked for between 45 and 180 min, after which it is cooled for 10–15 min before consumption. Relish is primarily cooked in the morning in large amounts ready for lunch so that it can be eaten during lunch, dinner and, sometimes, on the following day. Relish samples were collected at three main times: at lunch after cooking and after serving in a container (mostly plate); at dinner from a storage container, after reheating and after serving; and the following morning from a storage container, after reheating and after serving.

Another set of environmental samples were taken from utensils. Utensil samples were taken using swabs from plates before serving the relish, nsima and porridge at lunch, at dinner and at breakfast the following morning. Spoons which were used by the child when eating porridge were also swabbed.

Food samples of approximately 200 g were taken using the household utensil that was used for serving or feeding the child porridge. The samples were placed in a sterile bag with a tight-fitting seal and stored in a cold box at a temperature of 4 °C. The samples were transported approximately 80 km from Chikwawa District to a microbiology laboratory at the College of Medicine (Blantyre) within 5 h of sampling. Each sample collected was accompanied with details of the time, location and type of sample, whether it was fresh or stored food, whether the food was covered or not, and the temperature of the food at the time of sampling. Digital thermometers were used to measure the food temperature at four points: immediately after cooking, after serving in the utensil, after storage (4–6 h at room temperature), and immediately after reheating. The number of diarrhoea episodes per household for the preceding 2 weeks, together with the presence of flies or animals, and hygiene practices (e.g., handwashing at critical times and washing of utensils) were recorded.

All swab samples from the utensils were taken with sterile cotton swabs, stored in a peptone-buffered solution, and then transported to the laboratory. A 10-fold dilution was made, and 1 mL of the dilution was then transferred onto three different 3M^TM^ Petrifilm^TM^ plates: 3M^TM^ Petrifilm^TM^
*E. coli*/Coliform Count Plate, which was specific for *E. coli*, one specific for *Salmonella* sp. and one specific for *Staphylococcus aureus*. The 3M^TM^ Petrifilm^TM^ Count Plates are a sample-ready, culture-medium system that contains Violet Red Bile (VRB) nutrients, a cold-water-soluble gelling agent, an indicator of glucuronidase activity (BCIG), and a tetrazolium indicator that facilitates colony enumeration. The 3M^TM^ Petrifilms^TM^ were then incubated at 37 °C for 24 h. The 3M^TM^ Petrifilm^TM^ (3M Sciences SA, Rivonia, Johannesburg, South Africa) was used to identify and count bacterial colonies using an indicator dye and a built-in grid. Presumptive *E. coli* colonies (blue colonies with associated gas bubbles) were cultured in tryptone water at 44 °C for 24 h and an Indole test was performed with Kovac’s reagent. *Staphylococcus aureus* colonies were confirmed by observing yellow coloration on mannitol salt agar after incubation for 24 h at 37 °C. *Salmonella* sp. was confirmed by growing colonies on XLD agar, and the resulting positive colonies were subcultured onto nutrient agar. For serological confirmation and serotyping of Salmonella, API^R^ 20E (BioMérieux^®^ SA, Johannesburg, South Africa) biochemical and serology tests were done with Poly O and Poly H antisera. Thick food solids (100 g each) were homogenized with 90 mL of sterile buffered peptone water and homogenized in a stomacher blender. A 10-fold dilution was made and processed as described above.

### 2.6. Data Analysis

Field notes from in-depth interviews conducted after observations were analysed to identify themes for each target behaviour. These were in line with study themes such as complementary feeding practices, and the willingness to change food hygiene related behaviours. Checklist and structured observation data were reviewed and summarised to identify food contamination pathways during food storage, preparation, child feeding, reheating, and handwashing with soap at critical times.

Quantitative household data collected through Open Data Kit software (ODK) were exported to Microsoft Excel and quality checked before being exported to SPSS (version 25), where demographics, socio-economic measures, child health status and hygiene proxy measures were summarised.

### 2.7. Identification of Critical Control Points

The study used a HACCP approach to identify the critical control points (CCP) based on observed practices for foods consumed by under five children. This structured approach used the data from the study area to consider microbiological hazards from raw food storage to consumption [47].

### 2.8. Ethics

Ethical approval for this study was received from the College of Medicine Research Ethics Committee (P.04/16/1935). The study was registered with the Pan African Clinical Trials Registry (PACTR201703002084166). Written, informed consent and assent was obtained from all caregivers of children participating in the study.

## 3. Results

### 3.1. Demographic Characteristics

Of the 323 respondents (primary caregiver of the target child), the majority (66%) were in the age range of 18–28 years (Table 2). The age range of targeted children was 6–24 months (mean 14.27 with SD 5.72), of which 51% were male. A majority (90%) of families were living below the extreme poverty line (less than USD 1.25 per day), which was reflected in the levels of education, occupations and standard of housing, as summarized in Table 2. No participating households were connected to an electrical power supply and none owned a refrigerator. Of the sampled population, 95% had a latrine, which was unsurprising as the area was declared Open Defaecation Free by the Ministry of Health in 2016. Nevertheless, the majority (65%) of latrines were unimproved, and only half of them had a drop-hole cover. Despite this, only 4% of latrines had observable faeces around the drop hole. A specific place for handwashing, mostly being tippy taps (37%) was found in 51% of households. However, only 19% of handwashing facilities had soap and water. We found that the majority (64%) of handwashing facilities were located near the latrine; again, indicative of the recent Community-Led Total Sanitation campaign in the area. However, more traditional handwashing facilities such as basins and jugs were available in the household yard, and were observed to be more accessible for handwashing during food preparation and before eating (Table 2).

Animal ownership in the area was high (65%), with the majority of these being small domesticated animals such as chickens and goats who resided both inside and outside the house. As such, animal faeces was evident in 53% of the household yards. We found that 64% of the households kept their domestic animals inside the house at night for security and the houses had no separate room for keeping animals.

### 3.2. Food and Hygiene Proxies

We collected both self-reported and observed data on the children’s food and feeding practices (Table 3). Children were likely to start solid foods under the recommended age of 6 months (40%), although the majority were still breastfed, regardless of their age (87%). Children were given a range of foods to eat, with the majority receiving maize-based porridge (94%), and over half eating the same foods as the rest of the family at lunch and supper, which was comprised of nsima and relish (e.g., beans, vegetables). Children primarily ate at home but ate with others such as neighbours or relatives (89%). In all locations, children ate either on the veranda or ground outside the house in direct contact with dirt (42%) or placed on a reed mat (58%). Utensils for cooking and eating were reported to be washed more often after use, rather than before use; some utensils had gathered visible dust because of prolonged storage after washing. Only 6% of the utensils were found to be washed within 2 h before use.

### 3.3. Observational Results

Supplementary to the self-reported and observed information during the survey, the checklist and structured observations provided more detailed insight to the hygiene practices around under- two caregiving. As shown in Table 4, caregivers did not wash their hands with soap at all of the opportunities observed before food preparation, after attending to animal faeces and before eating which included child feeding. From the in-depth interviews, it was learned that caregivers did not wash hands before food preparation because of lack of proper handwashing facilities nearby. One caregiver commented: “It’s very difficult to wash hands when preparing food because there is no handwashing facility nearby that can allow me to do so without assistance. Mostly if I am to wash hands then I use a cup, but I always need someone to pour water over my hands to wash properly. Unfortunately, in most cases I am only with the child.”

A lack of handwashing with soap during food preparation and eating/child feeding is related to the fact that there is rarely a specific place for handwashing in the household yard, and that a majority of the handwashing facilities are located close to the latrine (64%). Facilities for handwashing in the household yard, where most activities related to hygiene take place, were buckets without a tap, which made self-handwashing difficult. When asked why they did not use the tippy tap located near the latrine as an alternative, respondents stated that the tippy tap was too far and also it would be disgusting for them to use a handwashing facility near the latrine while preparing food or before eating. 61% of the households had soap, but only 19% placed the soap at the handwashing station. During IDIs with the caregivers, they reported that soap was expensive ($0.20 per bar); hence, it is prioritized for washing clothes and bathing.

Results noted during checklist observations were similarly observed during structured observations, where the majority of caregivers did not wash hands with soap at critical times (Table 5). Nevertheless, all adults practised what they called handwashing before eating main meals. However, none of the adults washed their hands with soap, and 63% of them dipped their hands in one communal bowl or pot of water for a few seconds as a means of washing. During an in-depth interview, one caregiver commented that: “Eating nsima without handwashing is something we consider abnormal in this village … and I do not feel comfortable eating nsima without washing hands because it sticks in the hands … and everyone washes hands in our family before eating nsima.”

Children were also seen mouthing a variety of objects during the observation periods (Table 6). These items included hands (their own, siblings’ and mothers’), inanimate objects such as cloth, maize cobs, shoes, stones, sticks, phones, utensils, paper, animal faeces and toys. They were also seen eating soil directly. Although over 90% of caregivers indicated that they monitor and prevent their children from mouthing dirty objects, we observed that the caregivers could not monitor children all the time, as they were sometimes busy with other household chores (e.g., cooking and collecting water).

Children were observed to eat the reported range of foods, with the main meals consisting of porridge, relish and nsima, with snacks including local fruits (e.g., cucumbers, mangoes, etc.) and commercial foods (e.g., maize puffs). Like adults, children washed their hands before taking their main meals by dipping their hands in one communal bowl. However, we did not observe any hand-washing before eating snacks. Forty-two percent of children were observed to self-feed, 30% were fed with a spoon by the caregiver (who, in 48% of cases, shared the utensil) and 25% were fed using the caregiver’s hand. When children self-fed with a spoon, it was observed to fall on the ground, and continued to be used without any washing.

During food preparation, opportunities for cross contamination were noted, including the lack of handwashing, and multi-tasking during cooking. For example, caregivers were seen to change a child’s nappy or remove mucous from the child’s nose while cooking, then resume food preparation without washing their hands. Once the food was prepared, 48% of households covered cooked foods prior to consumption. However, 19% of households were seen to leave a child’s porridge uncovered to allow it to cool before consumption, leaving it open to flies and animals in the vicinity.

Up to 55% of households were observed to keep leftover food stored for the next meal which could be between 1 and 18 h later. Leftovers were primarily the children’s porridge (11%), which was consumed shortly after preparation as it was left either to cool, or until the child was awake or not fussing; relish (43%), which was made of a combination of either green leaves, tomatoes, onions, or beans; and nsima (18%), which was eaten at the next meal. Bean-based relish was the most commonly stored food due to its long cooking time (about 3 h). Thus, caregivers preferred to cook relish once while nsima, which is quicker to cook (40 min), was prepared twice a day. Forty-five percent of households were observed to reheat leftover food, predominantly relish, as it was reported that reheated food tastes better than cold food. One caregiver commented during an IDI that: “We are always busy, so it is difficult and tiresome to cook the same type of relish more than once in a day … we just cook once to be enough for lunch and dinner and sometimes for breakfast for children on the following day especially if we would go to the agriculture field … also, firewood is very scarce here; hence, cooking at once saves firewood.”

Twenty-one percent of children defaecated during observations. Defaecation always took place in the household yard; all of the faeces was removed from the immediate vicinity, and 76% was disposed of in the toilet. The remainder was thrown into the bushes around the household. Animal faeces was observed in 66% of the household yards. From in-depth interviews, we noted that the caregivers did not pay much attention to animal faeces, as they considered it less harmful than human faeces. One caregiver reported: “We do not bother removing animal faeces as it is not very dangerous compared to human faeces … in fact, it is a good source of manure; hence, we just throw it in the garden when sweeping the household yard in the morning.”

### 3.4. Microbiological Results

As shown in Table 7, 224 microbiological samples were collected from 20 households, sampled at 3 different points; breakfast (*n* = 116), lunch (*n* = 38) and dinner (*n* = 70). We found that 30% of children within the sampled households had suffered from diarrhoea in the 2 weeks preceding, which was consistent with responses from the household survey (27%). The lack of a drop hole cover on latrines (50%), and the presence of animal faeces around the eating area (49%), in combination with the flies observed during food preparation and consumption (51%), raised concerns regarding their potential role in faecal-oral pathogen transmission in the area. 

Generally, porridge was produced for immediate consumption, with leftovers being kept on only 3 occasions in the sampled households, which aligns with reported practice in the survey. All leftovers were stored in the pot in which the porridge had been cooked and left on the ground with a plate to cover it. Relish was produced predominantly for lunch (100%) and was then used again for the evening meal or breakfast (73%), meaning that these foods had the longest storage time at ambient temperature. Of all relish stored, 96% was stored in a pot or plate, of which 89% was covered. Seventy-six percent of stored food was reheated to an average temperature of 53 °C. Nsima was cooked fresh twice a day: at lunch and again for supper. Leftover nsima was stored overnight in pots and plates, with 92% being covered with a plate and 84% being placed on the ground. Eighty-seven percent of households reheated nsima for consumption at breakfast to an average temperature of 52 °C. No foods were visibly spoiled at the time of sampling.

Both total coliforms and faecal coliforms showed a significant increase in food stored for over 2 h (Figure 2). This was particularly evident in the storage of relish, which was produced at lunch on Day 1 and consumed in the morning of Day 2 as part of breakfast, with an average storage time of 18 h.

Relish is reheated twice in a typical day: once for dinner, and once again for breakfast the next morning. However, an increase in the concentration of total and faecal coliforms was observed as the relish storage duration was prolonged (Figure 3a). Though the temperature does not strongly predict the presence or concentration of total coliforms, faecal coliforms appear in nsima that has been stored through the night, and the concentration is reduced by an increased serving temperature (Figure 3b). It is important to note that although reheating took place in practice, food was only reheated to the recommended 70 °C on 7 occasions (6%). We did not measure the period of time over which the temperatures were achieved, and as such, the reheating process should be examined in more detail to determine if an effective time and temperature combination can be reached taking into consideration barriers to this practice including time and cost. Of particular concern was the identification of *Staphylococcus aureus* in stored food samples. These results are indicative of poor hygiene practice related to household handwashing, and of concern in stored foods due to their production of heat stable toxins which are not destroyed by normal cooking (reheating) temperatures.

Freshly prepared nsima contained both total and faecal coliforms, and when the temperature dropped down to ambient temperature during storage, there was an increase in total coliforms and faecal coliforms (Figure 3b). When the nsima was subsequently reheated up to over 50 °C, all faecal coliforms were killed, but some coliforms remained, essentially unchanged from the initial product. Nsima is solid when cold, and reheating it to a consistent temperature throughout can be difficult and time consuming. Faecal contamination in this case is likely to be caused by poor handling of utensils and poor hand hygiene. As storage containers were reported to be covered, contamination was likely to be on the surface of the food, and therefore more easily destroyed during reheating.

In all cases, the cleanliness of the utensils and containers was an important variable. Although the majority of caregivers (75%) reported that they used soap when washing utensils, less than a third (29%) were observed using the soap. Alternatively, caregivers were observed to use sand/soil (53%), which could be contaminated with animal faeces. In addition, utensils were left on the ground and in areas where animals could access them. In some households (46%), animals were observed licking dirty utensils placed in a bucket or drinking water meant for cleaning. Microbiology results (Table 8) showed coliform contamination but an absence of faecal organisms.

### 3.5. Hazard Analysis

Based on the results of the qualitative and quantitative data analysis, the preparation of porridge (complementary food) and other family meals (nsima and relish) were visualized as process flow diagrams and subject to a risk assessment based on the Hazard Analysis Critical Control Point (HACCP) approach. The resultant flow diagrams (Figure 4a,b) summarize the methods of preparation while highlighting the key risks to contamination and the associated critical control points. Both figures describe the risk factors and critical control points for porridge as well as relish and nsima.

Referring to Figure 4a, the CCPs for the main complementary food (porridge) were cooking, implying that the cooking temperature should be adequate (i.e., 75 °C+); cooling should be achieved quickly and food should not be accessed by animals or flies. Children should be fed with clean utensils after the caregiver has washed her/his hands with soap. CCPs for nsima and relish (Figure 4b) were similar to porridge (i.e., cooking, cooling, and feeding the child). Furthermore, since the nsima and relish are stored to be eaten during the next meal, the additional CCPs included safe storage of food (controlled storage time and temperature; food must be covered) and reheating (up to boiling) before consumption. All datasets are available as Appendix A-link.

## 4. Discussion

Our results show that 27% of the children had suffered from diarrhoea two weeks prior to the study, which was 5% higher than the national childhood diarrhoea prevalence reported by the Malawian Demographics and Health survey [2]. Such an increased rate of diarrhoea, compounded by low levels of subsistence living requires further attention and prevention strategies in this rural setting of Malawi.

### 4.1. Household Meals and Contamination

Observations of complementary and family meals showed a relatively homogenous diet across the studied population. Foods were simple in nature and preparation, and in the case of children’s porridge, had low levels of contamination due to the short storage/cooling times and immediate consumption. As such, the critical control points relate to the potential post-cooking contamination sources such as hands, utensils and flies/animals. Family meals of nsima and relish were more complex and leftovers were frequently stored for consumption later the same day or the next morning. As such, preventing food contamination before and during storage, along with controlling the temperature of leftover food, are critical to ensure that pathogens cannot multiply and/or are killed prior to consumption. In the absence of a cold chain in this setting, it is therefore imperative that leftover foods be a focus of food safety interventions that support high hygiene standards commensurate with the environment, such as storing food in clean and sealable containers for limited time periods.

Although the measured levels of porridge contamination were lower than those reported in Nepal, comparably low levels of contamination in complementary foods were reported in a similar rural area of Zimbabwe, where mothers were also the primary caregivers. The similarities may indicate normative regional practices in child feeding [14,40]. Despite the fact that the complementary food is safe for consumption, the method and environment in which children were being fed were risky. For instance, the practice of placing children directly on the ground during feeding increased the risk of contamination. In similar settings in Zimbabwe and Tanzania, soil analysis found *E. coli* to be ubiquitous around the household yard which could be inadvertently consumed by children during feeding, mouthing and direct consumption as we observed in this study [14,22]. Furthermore, the study in Zimbabwe estimated that a one-year old child could consume up to 4,700,000 *E. coli* counts per day, a result which could be compared to this study’s setting due to the ubiquitous nature of animal and child defecation in the household yard [14].

### 4.2. Storage and Reheating of Food

Storing food overnight was a common practice, as caregivers were primarily subsistence farmers, and needed to save both time and fuel by preparing labour-intensive foods in the morning. Though food was adequately heated during cooking, the long storage time provided a conducive environment for microbial growth and multiplication. Reheating left-over food reduced coliforms and faecal coliforms. However, not all foods were reheated before consumption (45% reheated) (Table 3), and the temperatures reached during reheating were not always sufficient to achieve complete die-off of thermo-tolerant organisms (only 6% of samples reached the recommended 70 °C). Inadequate food reheating could be attributed to the fact that the caregivers reheated the food with the motive of improving taste rather than to kill pathogens. A study conducted in Mali in which foods were reheated to temperature in excess of 90 °C showed full die-off of thermo-tolerant bacteria [38]. This may be a reflection of the type of food being heated by households in Malawi, as the Mali study had a thinner porridge and fish soup for reheating, and is also indicative of the need to understand the context in which the food is being prepared and reheated. Ninety-four percent of participants live below the extreme poverty line, and as such, they struggle to access firewood for reheating, and even if they can, have little time to reheat food thoroughly before consumption when there are competing tasks such as collecting water, attending to children and agriculture fieldwork. As the majority of family foods are cooked to a high temperature for long periods, contamination is minimal after preparation. Therefore, focus should be on minimizing post-cooking contamination and safe storage.

### 4.3. Handwashing Practice

Handwashing after faecal contact, before food preparation, and before child feeding/eating snacks was rare, but comparable to previous studies [32], universal handwashing only occurred when the whole family was eating lunch or supper. This practice is therefore instilled as a social norm, with no need for prompts to make it happen. Nevertheless, the quality of handwashing before eating and at other critical times was ineffective in most cases, with little to no use of soap and use of communal water for dipping hands thereby leading to further contamination. Leftover food from communal eating is therefore subject to not only faecal-oral contamination, but also Staphylococcal pathogens such as *Staphylococcus aureus*, which, given the opportunity to multiply at the storage temperatures recorded, will produce heat-stable toxins that will survive the reheating process and cause vomiting and diarrhoea.

As with the storage and reheating of food, we must be cognizant of the context in which respondents are washing their hands and the behavioural and structural barriers which may be influencing these practices. As such, handwashing promotion needs to address the appropriate location of handwashing facilities, issues of soap use, which, due to poverty, is prioritized for other domestic activities such as bathing and washing clothes. Elsewhere, we reported that caregivers do not see the benefit in using soap for handwashing as they see no direct link between use of soap and a reduction of diarrhoeal disease in children [46]. As such, promoting handwashing with soap at critical times needs not only the provision of infrastructure, but also the development of effective behaviour-centred health promotion strategies.

### 4.4. Management of Household Utensils

Although this study did not show utensils to be contaminated with faecal pathogens, we observed that utensils (both clean and dirty) were left on the ground or in the open for long periods of time over which they could become contaminated with dust, faeces or from roaming animals. Previous research has reported the contamination of plastic plates and cups in Tanzania, and found high levels of faecal organisms in kitchen settings similar to those observed in this study [22]. As such, the role of utensils and the environment in which they are stored and used should be considered as a potential route of transmission for faecal-oral infections, we would recommend that items be washed just prior to use to minimise the risk of cross contamination. In addition, utensils should be rinsed with soap and water if sand was initially used to remove heavy stains. 

### 4.5. Limitations and Further Research

This study has several limitations. Collection of study samples was conducted in October 2017 during the hot, dry season in Chikwawa, which has an average temperature of 29 °C [48]. As such, contamination levels of food and hands may be higher than would be expected in the cooler season since high temperatures favour microbial growth and survival. Further studies assessing the microbiological quality of complementary foods in both summer and winter seasons are necessary. Observations at each household were conducted by a single research assistant which increased the burden of recording events to one person and may have led to a lack of detail in some reports where concurrent activities occurred. However, as complementary food hygiene is continuously being promoted, research findings such as ours may provide guidance to public health programme designers to develop effective food hygiene promotion strategies. Although undertaken within a larger research study, of which this study is a component, water quality in households was not tested within this formative population. The study would have benefited from microbiological sampling of household members’ hands after washing prior to the main meal to determine the efficacy of their practice. Hence, microbiological examination of the household water for drinking, cleaning utensils and handwashing would be an important component of future research.

## 5. Conclusions

This study examines the risk factors for faecal-oral routes of infection for children under the age of two years in rural Malawi. The results indicate that complementary foods produced solely for the child are relatively free from contamination, though there is a high risk associated with shared family meals, particularly those prepared from leftovers. Risks were identified from poor hand hygiene at critical times, e.g., after faecal contact, before food preparation, and before child feeding. Although handwashing before family meals was universal, the method was poor. Our findings also concur with previous studies showing that children are at risk from faecal-oral infection from their continuous contact and consumption of contaminated soil both directly and indirectly.

Interventions to reduce the risk factors should focus on the critical control points in food preparation, storage and reheating, and the contributing factors to post-cooking contamination such as hand hygiene, clean utensils and reducing contact with flies and animals. Interventions should respond to the contextual needs in which the practices occur and should be based on a behaviour-centred approach to create social norms around appropriate motives.

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
