# Peer review of "Risk Factors Associated with Feeding Children under 2 Years in Rural Malawi—A Formative Study"

_ijerph, 2019, doi:10.3390/ijerph16122146_

Round 1

Reviewer 1 Report

The paper is scientifically sound, study design and overall presentation are also fine. The main concern here however is the novelty of the study. Authors need to clarify the rational of the study better in the introduction. The sources of diarrhea is now well-known, what remains to be done is to explore how it can be prevented.  

There is no statistics on diarrhea in Malawi in the introduction. The text is very general and could fit in almost any diarrhea related  report. Since the study is on Malawi, readers expect to learn the country specific contexts rather than the global scenario. 

After a lot of search I am still failing to find the year of data collection. It should be mentioned clearly in the abstract. 

In the discussion, the subject matter of the study demands some real-life interpretation of the findings. To whom the results are directed, how the results can be used to improve the situation? 

A clear and concise take home message should to made at the end of discussion and conclusion. 

Author Response

Response to reviewer number 1 have been uploaded.

Reviewer 2 Report

This paper examines food safety preparation and storage practices for children under 2 years old in rural Malawi to improve household health using a mixed methods approach. 

Abstract

1.)  Change “structured (n=80) observations” to “structured observations (n=80),”

2.)  Line 24, break this sentence into two

3.)  Line 30, change to “handwashing facilities”

Add food safety as a keyword. 

Main text

1.)  Replace the Fischer Walker (2013) reference with recent WHO statistics globally and for Malawi found at https://www.who.int/gho/child_health/mortality/en/

2.)  Consider the findings of the following reference “Fecal Contamination and Diarrheal Pathogens on Surfaces and in Soils among Tanzanian Households with and without Improved Sanitation” by Amy J. Pickering et al. Compare the results of your study to this work in Tanzania. 

3.)  Line 72, consider adding something about the importance of local cultural and practices that affect this food safety situation which make cross cultural studies not able to be compared. 

4.)  Line 101 to 104, this would be better presented in a simple table. 

5.)  Please provide more detail on the random sampling, or was it convenience sampling?

6.)  Line 120, again more details on random sampling are needed. 

7.)  Line 158, more details on how these random households were selected is needed. 

8.)  Line 161, please define “Relish” and “Nsima” and/or add a picture of these. 

9.)  Did you use Ringers solution for the swabs or was it a swab just put in water?

10.)                 Table 2, snacks such as Kamba are very common for children in Malawi, and it is surprising these are not included in this table. 

11.)                 Table 2, When are utensils washed (n=323), please reorder this to start with before eating then ending >2 hours after eating.

12.)                 Line 285. Add the price of local soap to this section. 

13.)                 Line 297 and 318, be clear that respondents are calling this handwashing but in reality it is dipping hands in a communal bowl of water for a few seconds. 

14.)                 Line 334, what was the estimated time to cook beans?  Then compare this to the time to cook Nsima.  For example, beans take a long time to cook so respondents only cooked once per day whereas Nsima is faster to cook so they made it twice per day. 

15.)                 Table 6, add “Celsius” to temperature column 

16.)                 Figure 2 and 3a, add approximate numbers of hours this figure covers to the legend. 

17.)                 It can be worth noting that despite laboratory analysis, food did not visually appear to be spoiled during sampling if this was the case. 

18.)                 Line 454, this is the first mention of absence of cold chain.  Consider adding this earlier, that for example no refrigerators were at the homes of respondents if that was the case. 

19.)                 Line 525, the study limitations section is well written.  Considering adding 2 to 4 future research topics. 

General comments

1.)  There is a mix of British and American spellings, of diarrheal and fecal, please just double check. 

2.)  Change throughout “hand washing” should be “handwashing”

3.)  Finally, when you submit the corrected version, please do check thoroughly, in order to avoid grammar, syntax or structure/presentation flaws - please seek for professional English proofreading services or ask a native English-speaking colleague of yours in order to refine and improve the English in your paper.

Author Response

Response to comments from reviewer number 2 have been uploaded.
